# GraphEBM: Towards Permutation Invariant and Multi-Objective Molecular Graph Generation

## Abstract

Although significant progress has been made in molecular graph generation recently, permutation invariance and multi-objective generation remain to be important but challenging goals to achieve. In this work, we propose GraphEBM, a molecular graph generation method via energy-based models (EBMs), as an exploratory work to perform permutation invariant and multi-objective molecule generation. Particularly, thanks to the flexibility of EBMs and our parameterized permutation-invariant energy function, our GraphEBM can define a permutation invariant distribution over molecular graphs. We learn the energy function by contrastive divergence and generate samples by Langevin dynamics. In addition, to generate molecules with a specific desirable property, we propose a simple yet effective learning strategy, which pushes down energies with flexible degrees according to the properties of corresponding molecules. Further, we explore to use our GraphEBM for generating molecules towards multiple objectives via compositional generation, which is practically desired in drug discovery. We conduct comprehensive experiments on random, single-objective, and multi-objective molecule generation tasks. The results demonstrate our method is effective.

## 1 Introduction

A fundamental problem in drug discovery and material science is to find novel molecules with desirable properties. One popular method is to search in the chemical space based on molecular property prediction (Gilmer et al., 2017; Wu et al., 2018; Yang et al., 2019; Stokes et al., 2020; Wang et al., 2020). In addition, molecular graph generation provides an alternative and promising way for this problem by directly generating desirable molecules, thus circumventing the expensive search of the chemical space. As reviewed in Section 2.1, existing approaches have achieved promising success by generating molecular graphs based on various generative methods, including variational autoencoders (VAEs) (Kingma & Welling, 2013), generative adversarial networks (GANs) (Goodfellow et al., 2014), flow models (Dinh et al., 2014; Rezende & Mohamed, 2015) and recurrent neural networks (RNNs). Despite these intensive efforts, we still face the following significant challenges.

**(i) Data distribution is permutation invariant.** Permutation invariance is known as an intrinsic and desirable inductive bias for graph data. If we permute the node order of a given molecular graph, the resulting graph still corresponds to the same molecule. In other words, the underlying data distribution of molecular graphs is invariant to permutations. Thus, an ideal generative model, which aims to capture the underlying data distribution, should associate the same probability to various permutations of the same molecular graph (Niu et al., 2020). However, as analyzed in Section 2.1, most existing methods fail to preserve the intrinsic property of permutation invariance during density modeling.

**(ii) Multi-objective generation is in demand but challenging.** An ultimate goal of drug discovery is to obtain molecules that have multiple properties simultaneously (Jin et al., 2020b). Unfortunately, we usually do not have a training set of enough molecules that satisfy multiple desired properties simultaneously. In other words, if we have such dataset, there is no need to generate more molecules since we already have enough desired molecules in the dataset. Hence, it is greatly desired but challenging to generate molecules with multiple objectives.

We are the first to observe that developing molecular graph generative model based on energy-based models (EBMs) (LeCun et al., 2006) has the potential to perform permutation invariant and multi-objective molecular graph generation. In this study, we propose GraphEBM to explore permutation invariant and multi-objective molecular graph generation. Since we use the flexible EBM framework and a permutation invariant energy function, our GraphEBM can provide a permutation invariant distribution over molecular graphs, thus preserving the intrinsic permutation invariance property during density modeling. We apply contrastive divergence (Hinton, 2002) to learn the energy function and generate samples from it via Langevin dynamics (Welling & Teh, 2011). To our knowledge, our GraphEBM is the first energy-based model that can generate attributed molecular graphs. To achieve multi-objective generation, we first need to generate molecules with a single objective property. Particularly, we propose a novel, simple, and effective strategy to train our GraphEBM for single-objective generation by pushing down energies with flexible degrees according to the property values of corresponding molecules. Further, we propose that GraphEBM can generate molecules with multiple objectives in a compositional manner. To be specific, we first learn multiple single-objective EBMs since the training set for a single objective is usually available. Afterwards, we can perform multi-objective generation via the compositionality endowed by EBMs. This provides a new and promising way for multi-objective molecule generation, which is significantly helpful for drug discovery.

**Contributions.** (i) We propose the first EBM that is capable of generating molecular graphs with preserving permutation invariance property. (ii) To achieve single-objective generation with GraphEBM, we propose to push down energies with flexible degrees according to the corresponding property values, which is simple and effective. (iii) Based on single-objective generation, we demonstrate that our GraphEBM can perform multi-objective molecular graph generation compositionally, which provides a novel and promising way for multi-objective molecule generation. (iv) We empirically demonstrate that our approach is effective via random, single-objective, and multi-objective molecule generation.

## 2 PRELIMINARY AND RELATED WORK

### 2.1 MOLECULAR GRAPH GENERATION

Since molecules can be represented as SMILES strings (Weininger, 1988), early studies generate molecules based on SMILES strings, such as CVAE (Gómez-Bombarelli et al., 2018), GVAE (Kusner et al., 2017), and SD-VAE (Dai et al., 2018). Recent studies mostly represent and generate molecules as graphs (Simonovsky & Komodakis, 2018; De Cao & Kipf, 2018; Madhawa et al., 2019). We can categorize existing molecular graph generation methods based on the underlying generative methods or the generation processes. Current molecular graph generation approaches can be grouped into four categories according to their underlying generative models, *i.e.*, VAEs, GANs, flow models, and RNNs. They can also be classified into two primary types based on their generation processes; those are, sequential generation and one-shot generation. The sequential process generates nodes and edges in a sequential order by adding nodes and edges one by one. The one-shot process generates all nodes and edges at one time.

To facilitate comparison, we summarize existing methods in Table 1. We can observe that most of them fail to satisfy an intrinsic property of graphs; that is, permutation invariance. Specifically, a generative model should ideally yield the same likelihood for different permutations of the same graph, since the underlying data distribution is permutation invariant. However, the sequential generation approaches model graphs by choosing a specific order of nodes, thus failing to preserve permutation invariance. Among the one-shot methods, Bresson & Laurent (2019) also use the specific node order given by the SMILES representation. GraphVAE and RVAE perform an approximate and expensive graph matching to train the VAE model, and they cannot achieve exact permutation invariance. MolGAN circumvents this issue by using a likelihood-free method. The recent one-shot flow methods have the potential to satisfy this property. However, GraphNVP, GRF, and MoFlow cannot preserve this property since the masking strategies in the coupling layers are sensitive to node order. An exception is GraphCNF, which achieves permutation invariance by assigning likelihood independent of node ordering via categorical normalizing flows. Furthermore, the existing methods, shown in Table 1, cannot easily achieve multi-objective generation for practical use. Compared with these existing works, we propose to develop EBMs for molecular graph generation. We note that

Table 1: Summary and comparison of existing molecular graph generation methods.

| Method | Generative method | | | | | Generation process | | Permutation invariance | Multi-objective generation |
|---|---|---|---|---|---|---|---|---|---|
| | *VAE* | *GAN* | *Flow* | *RNN* | *EBM* | *One-shot* | *Sequential* | | |
| GraphVAE (Simonovsky & Komodakis, 2018) | ✓ | - | - | - | - | ✓ | - | ✗ | - |
| DeepGMG (Li et al., 2018) | - | - | - | ✓ | - | - | ✓ | ✗ | - |
| CGVAE (Liu et al., 2018) | ✓ | - | - | - | - | - | ✓ | ✗ | - |
| MolGAN (De Cao & Kipf, 2018) | - | ✓ | - | - | - | ✓ | - | - | - |
| RVAE (Ma et al., 2018) | ✓ | - | - | - | - | ✓ | - | ✗ | - |
| GCPN (You et al., 2018) | - | ✓ | - | - | - | - | ✓ | ✗ | - |
| JT-VAE (Jin et al., 2018) | ✓ | - | - | - | - | - | ✓ | ✗ | - |
| MolecularRNN (Popova et al., 2019) | - | - | - | ✓ | - | - | ✓ | ✗ | - |
| GraphNVP (Madhawa et al., 2019) | - | - | ✓ | - | - | ✓ | - | ✗ | - |
| Bresson and Laurent (Bresson & Laurent, 2019) | ✓ | - | - | - | - | ✓ | - | ✗ | - |
| GRF (Honda et al., 2019) | - | - | ✓ | - | - | ✓ | - | ✗ | - |
| GraphAF (Shi et al., 2019) | - | - | ✓ | - | - | - | ✓ | ✗ | - |
| HierVAE (Jin et al., 2020a) | ✓ | - | - | - | - | - | ✓ | ✗ | - |
| MoFlow (Zang & Wang, 2020) | - | - | ✓ | - | - | ✓ | - | ✗ | - |
| GraphCNF (Lippe & Gavves, 2020) | - | - | ✓ | - | - | ✓ | - | ✓ | - |
| GraphDF (Luo et al., 2021) | - | - | ✓ | - | - | - | ✓ | ✗ | - |
| **GraphEBM** | - | - | - | - | ✓ | ✓ | - | ✓ | ✓ |

EBMs have unique advantages for molecular graph generation, including preserving permutation invariance and enabling multi-objective generation via compositionality.

## 2.2 ENERGY-BASED MODELS

Modeling variables by defining an unnormalized probability density has been explored for decades (Hopfield, 1982; Ackley et al., 1985; Cipra, 1987; Dayan et al., 1995; Zhu et al., 1998; Hinton, 2012). Such methods are known as energy-based models (EBMs) (LeCun et al., 2006) in machine learning. Given a data point $x$, let $E_\theta(x) \in \mathbb{R}$ be the corresponding energy, where $\theta$ denotes the learnable parameters of the energy function. Then, the energy function defines a distribution as

$$p_\theta(x) = \frac{e^{-E_\theta(x)}}{Z(\theta)} \propto e^{-E_\theta(x)}, \qquad (1)$$

where $Z(\theta) = \int e^{-E_\theta(x)} dx$ is the normalization constant and usually intractable. Currently, EBMs have been used as generative models in multiple domains, including images (Xie et al., 2015; 2016; Du & Mordatch, 2019; Du et al., 2020a;b), videos (Xie et al., 2017), texts (Deng et al., 2020), 3D objects (Xie et al., 2018), point sets (Xie et al., 2020), and proteins (Du et al., 2020c).

To date, EBMs have rarely been studied in the graph domain. Liu et al. (2020) attempts to generate graph structures by building EBMs based on graph neural networks. However, it can only generate graph structures, *i.e.*, binary adjacency matrices. It is not straightforward to extend for attributed graphs since the number of data dimensions of attributed graphs is much larger than graphs without attributes, thereby leading to difficulty for discrete sampling method used in Liu et al. (2020) to draw samples from model distribution. In addition, they do not consider objective-directed graph generation. We note that there is a concurrent work (Hataya et al., 2021) with ours, which also generate molecules with EBMs. It focuses on substructure-preserving molecule generation, while we emphasize the advantage of permutation invariance endowed by EBMs for generating molecules from scratch. In addition, we further investigate multi-objective molecule generation, which is practically desired but unexplored in Hataya et al. (2021).

## 3 THE PROPOSED GRAPHEBM

In this section, we present GraphEBM by describing our parameterized energy function (Section 3.2), showing that GraphEBM achieves permutation invariant generation (Section 3.3), and describing the learning (Section 3.4) and generation (Section 3.5) process of GraphEBM. Then, we introduce our proposed strategy for single-objective generation based on GraphEBM (Section 3.6). Finally, we explain how to use GraphEBM for multi-objective molecule generation via compositional generation (Section 3.7).

### 3.1 PROBLEM FORMULATION

Molecules can be naturally represented as graphs by considering atoms and bonds as nodes and edges, respectively. We formally represent a molecular graph as $\mathcal{G} = (X, A)$, where $X$ is the node

feature matrix and $A$ is the adjacency tensor. Let $k$ be the number of nodes in the graph. $b$ and $c$ denote the number of possible types of nodes and edges, respectively. Then we have $X \in \{0, 1\}^{k \times b}$ and $X_{(i,p)} = 1$ if node $i$ belongs to type $p$. $A \in \{0, 1\}^{k \times k \times c}$ and $A_{(i,j,q)} = 1$ denotes that an edge with type $q$ exists between node $i$ and node $j$. Following prior works (Madhawa et al., 2019; Zang & Wang, 2020), we let $n$ denote the maximum number of atoms that a molecule has in a given dataset. We insert virtual nodes into molecular graphs that have less than $n$ nodes such that the dimensions of $X$ and $A$ keep the same for all molecules. Also, for any two nodes that are not connected in the molecule, we add a virtual edge between them. We can consider the virtual node and the virtual edge as an additional node type and edge type, respectively. Hence, for all molecules in a certain dataset, $X \in \{0, 1\}^{n \times (b+1)}$ and $A \in \{0, 1\}^{n \times n \times (c+1)}$.

## 3.2 PARAMETERIZED ENERGY FUNCTION

Following the above notations, the energy function for molecular graphs can be denoted as $E_\theta(X, A)$. In this work, we model $E_\theta(X, A)$ by a graph neural network, where $\theta$ denotes parameters in the network. Particularly, we use a variant of relational graph convolutional networks (R-GCN) (Schlichtkrull et al., 2018) to learn the node representations since molecular graphs have categorical edge types. Formally, the layer-wise forward-propagation is defined as

$$H^{\ell+1} = \sigma \left( \sum_{k=1}^{c+1} \left( A_{(:,:,k)} H^\ell W_k^\ell \right) \right). \tag{2}$$

$A_{(:,:,k)}$ is the $k$-th channel of the adjacency tensor. $H^\ell \in \mathbb{R}^{n \times d_\ell}$ is the node representation matrix at layer $\ell$, where $d_\ell$ denotes the hidden dimension at layer $\ell$. $W_k^\ell \in \mathbb{R}^{d_\ell \times d_{\ell+1}}$ represents the trainable weight matrix for edge type $k$ at layer $\ell$. $\sigma(\cdot)$ denotes a non-linear activation function. The initial node representation matrix $H^0 = X$. In each layer, message passing is conducted among the nodes independently for each type of edge. Then, the information is integrated together by a sum operator. We stack $L$ such layers. Hence, the final node representation matrix is $H^L \in \mathbb{R}^{n \times d}$, where $d$ is the hidden dimension. Then, the representation of the whole graph can be derived by a readout function. In this work, we use the sum operator to compute the graph-level representation $h_G$ as $h_G = \sum_{i=1}^n H_{(i,:)}^L \in \mathbb{R}^d$. Finally, the scalar energy associated with the molecular graph can be obtained as $E = h_G^T W \in \mathbb{R}$, where $W \in \mathbb{R}^d$ is the trainable transformation.

## 3.3 PERMUTATION INVARIANT GENERATION

Naturally, a distribution of molecular graphs can be denoted as $p(X, A)$. Since the underlying data distribution of molecular graphs is invariant to permutations, as analyzed in Section 1, an ideal generative model should associate the same probability to various permutations of the same molecular graph. We show that our proposed GraphEBM satisfies this fundamental property. Specifically, each layer of our graph neural network in Eq. (2) is permutation equivariant. In addition, the readout operation is permutation invariant. Therefore, our parameterized energy function is permutation invariant, thus satisfying $E_\theta(X, A) = E_\theta(X^\pi, A^\pi)$, where $\pi$ denotes any permutation of node order. For simplicity, we use the superscript $\pi$ to denote that the corresponding matrix or tensor is arranged according to the node order given by $\pi$. Further, the energy function defines a distribution over data space according to Eq (1). Specifically, the likelihood is proportional to the exponential of the negative energy. Hence, we can obtain $p_\theta(X, A) = p_\theta(X^\pi, A^\pi)$. Thus, our GraphEBM can define a permutation invariant distribution over molecular graphs.

Although it seems easy for our GraphEBM to achieve permutation invariant generation, the underlying rationale is non-trivial. It is brought by the unique advantage of EBMs, and this is why most existing methods based on other generative methods, as shown in Table 1, cannot achieve permutation invariant generation. To be specific, EBMs have great flexibility since the energy function defines the distribution by unnormalized values. Thus, the distribution given by an EBM can be simply represented as Eq (1), which is crucial for showing our GraphEBM can achieve permutation invariant generation as above. In contrast, generative processes with a tractable likelihood, such as autoregressive models (Graves, 2013) which are commonly used for sequential molecule generation, have less flexibility, thus losing the permutation invariance when modeling graphs. For example, in autoregressive models, model distribution has to be factorized as a series of conditional distributions. Hence, a specific order of nodes has to be predetermined, thereby failing to providing a permutation invariant distribution for graphs.

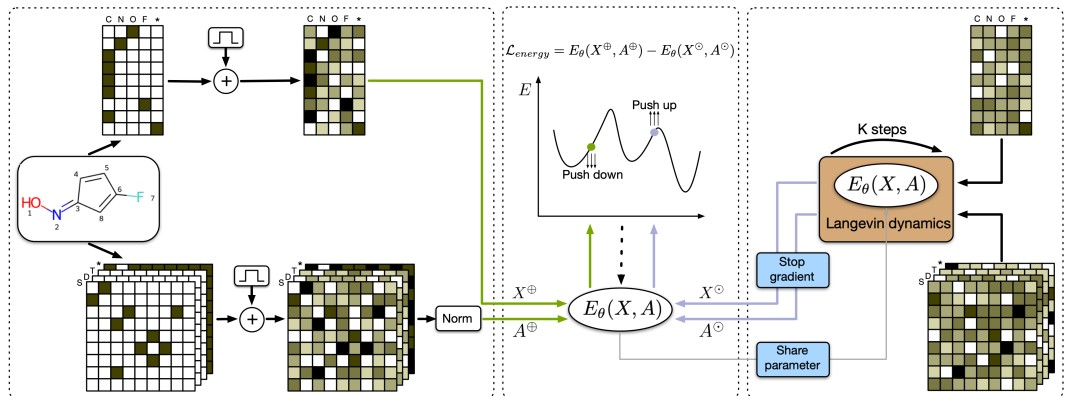

Figure 1: The learning process of our GraphEBM. The detailed annotation is in Appendix A.

## 3.4 LEARNING

Intuitively, a good energy function should assign lower energies to data points that correspond to real molecular graphs and higher energies to other data points. In this study, we apply contrastive divergence (Hinton, 2002) to learn the energy function.

Let $p_{\mathcal{D}}$ be the unknown distribution of the real data. To achieve maximum likelihood, we have $\mathcal{L}_{ML} = \mathbb{E}_{(X,A) \sim p_{\mathcal{D}}} [-\log p_\theta(X, A)]$, where $-\log p_\theta(X, A) = E_\theta(X, A) + \log Z(\theta)$ according to Eq. (1). It has been shown (Hinton, 2002; Turner, 2005; Song & Kingma, 2021) that this objective has the below gradient:

$$\nabla_\theta \mathcal{L}_{ML} = \mathbb{E}_{(X^\oplus, A^\oplus) \sim p_{\mathcal{D}}} \left[ \nabla_\theta E_\theta(X^\oplus, A^\oplus) \right] - \mathbb{E}_{(X^\odot, A^\odot) \sim p_\theta} \left[ \nabla_\theta E_\theta(X^\odot, A^\odot) \right]. \quad (3)$$

As defined in Eq. (1), $p_\theta$ is the distribution given by the energy function. Following Du et al. (2020b), we refer to $(X^\odot, A^\odot)$ as hallucinated samples. Intuitively, this gradient pushes down the energies of positive samples $(X^\oplus, A^\oplus)$ and pushes up the energies of hallucinated samples $(X^\odot, A^\odot)$ that are drawn from $p_\theta$. In practice, however, sampling $(X^\odot, A^\odot)$ from $p_\theta$ is challenging, since $Z(\theta)$ in Eq. (1) is intractable.

To overcome this issue, we follow Du & Mordatch (2019) to sample $(X^\odot, A^\odot)$ from an approximated $p_\theta$ using Langevin dynamics (Welling & Teh, 2011). Particularly, a sample $(X^\odot, A^\odot)$ is initialized randomly and refined iteratively by

$$X^k = X^{k-1} - \frac{\lambda}{2} \nabla_X E_\theta \left( X^{k-1}, A^{k-1} \right) + w^k,$$
$$A^k = A^{k-1} - \frac{\lambda}{2} \nabla_A E_\theta \left( X^{k-1}, A^{k-1} \right) + \eta^k, \quad (4)$$

where $w^k, \eta^k \sim \mathcal{N}(0, \sigma^2)$ are added noise sampled from a Gaussian distribution, $k$ denotes the iteration step, and $\frac{\lambda}{2}$ is the step size. As demonstrated by Welling & Teh (2011), the obtained samples $(X^k, A^k)$ approach samples from $p_\theta$ as $k \to \infty$ and $\frac{\lambda}{2} \to 0$. In practice, we let $K$ denote the number of iteration steps of Langevin dynamics and use the resulting sample $(X^K, A^K)$ as $(X^\odot, A^\odot)$ in Eq. (3).

We illustrate the training process of our GraphEBM in Figure 1. Since Langevin dynamics is for continuous data, we model the hallucinated samples by continuous format. For consistency, we can also use dequantization techniques (Dinh et al., 2016; Kingma & Dhariwal, 2018) to convert the discrete positive samples to continuous data by adding uniform noise[1], as shown in the left part of Figure 1. The dequantization can be formally expressed as

$$X' = X + tu, \ u \sim [0,1)^{n \times (b+1)}; \quad A' = A + tu, \ u \sim [0,1)^{n \times n \times (c+1)}. \quad (5)$$

$t \in [0, 1)$ is a scaling hyperparameter. After dequantization, we apply a normalization to the adjacency tensor. Formally,

$$A^\oplus_{(:,:,k)} = D^{-1} A'_{(:,:,k)}, \quad k = 1, \cdots, c+1, \quad (6)$$

---

[1]More discussion about dequantization is available in Appendix B

where $D$ is the diagonal degree matrix in which $D_{(i,i)} = \sum_{j,k} A'_{(i,j,k)}$. We treat the above $A^{\oplus}$ and $X^{\oplus} = X'$ as the input for the energy function. In our case, each element of $X^{\oplus}$ is in $[0, 1 + t)$ and each element of $A^{\oplus}$ is in $[0, 1)$.

To make the hallucinated sample $(X^{\odot}, A^{\odot})$ have the same value range as $(X^{\oplus}, A^{\oplus})$, we initialize it as $X^{\odot} \sim [0, 1 + t)^{n \times (b+1)}$ and $A^{\odot} \sim [0, 1)^{n \times n \times (c+1)}$. Then we apply $K$ steps of Langevin dynamics as Eq. (4) to refine the sample, as illustrated in the right part of Figure 1. After each step of refinement, we clamp the data to guarantee that the values are still in the desirable ranges.

As demonstrated in Eq. (3), the energies of positive samples are expected to be pushed down and the energies of hallucinated samples should be pushed up. Hence, to shape the energy function as expected, our loss function is defined as

$$\mathcal{L}_{energy} = E_\theta(X^{\oplus}, A^{\oplus}) - E_\theta(X^{\odot}, A^{\odot}). \qquad (7)$$

As shown in the middle part of Figure 1, the gradient backpropagated from $\mathcal{L}_{energy}$ can update the parameters $\theta$, thus pushing the energy function $E_\theta(X, A)$ to approach our expected shape. Notably, the gradient from $\mathcal{L}_{energy}$ will not be propagated to the energy function used in Langevin dynamics. We apply parameter sharing to keep the energy function used in Langevin dynamics up-to-date. The overall learning process of our GraphEBM is summarized in Algorithm 1, Appendix C.

### 3.5 GENERATION

Let $E_{\theta^*}(X, A)$ denote the learned energy function, where $\theta^*$ represents the obtained parameters. Intuitively, if an energy function is well-shaped, the configurations with low energies should correspond to desirable molecular graphs. Hence, the generation process is to generate molecules based on the configurations $(X, A)$ that yield low energies.

An overview of the generation process is shown in Figure 6, Appendix D. The steps are as follows. First, we initialize a data point as $(X^{\odot}, A^{\odot})$ and then apply $K$ steps of Langevin dynamics as in Eq. (4) to obtain data points that have low energies. We denote the obtained configuration as $(X^*, A^*)$. Second, since molecular graphs are undirected, we make the adjacency tensor to be symmetric by using $A^* + A^{*T}$ as the new adjacency tensor. Third, we convert the continuous data to discrete ones by applying the argmax operation in the dimensions of atom types and bond types. Finally, we use validity correction introduced by Zang & Wang (2020) to refine the corresponding molecule so that the valency constraint is satisfied.

### 3.6 SINGLE-OBJECTIVE GENERATION

For drug discovery and material design, we further need to generate molecules with a desirable chemical property. This task is termed as single-objective generation (*a.k.a.* goal-directed generation). As noted in Appendix E, it is not straightforward to apply existing strategies to our GraphEBM for single-objective generation. To generate molecules with desirable chemical properties, we propose a novel, simple, and effective strategy for learning our GraphEBM. Our basic idea is to push down energies with flexible degrees according to the property values of corresponding molecules. If a molecule has a higher value of desirable property, we push down the corresponding energy harder. Formally, for single-objective generation, the loss function defined in Eq. (7) becomes

$$\mathcal{L}_{energy} = f(y)E_\theta(X^{\oplus}, A^{\oplus}) - E_\theta(X^{\odot}, A^{\odot}), \qquad (8)$$

where $y \in [0, 1]$ is the normalized property value and $f(y) \in \mathbb{R}$ determines the degree of the push down. We use $f(y) = 1 + e^y$ in this work. Thus, energies of molecules with higher property values are pushed down harder. After training, the generation process is the same as described in Section 3.5.

### 3.7 MULTI-OBJECTIVE GENERATION

In addition to the single property constraint, it is more significant and desired to generate molecules with multiple property constraints in drug discovery. As analyzed in Section 1, multi-objective generation still remains challenging for current methods since we usually lack such datasets, in which molecules satisfy multiple constraints simultaneously. We observe that compositionality brought by EBMs (Hinton, 2002), which has been shown to be effective in the image domain (Du et al., 2020a), can be naturally applied to generate molecules with multiple constraints based on our GraphEBM.

Table 2: Generation performance on QM9.

| Method | Validity(%) | Uniqueness(%) | Novelty(%) |
|---|---|---|---|
| CVAE | 10.30 | 67.50 | 90.00 |
| GVAE | 60.20 | 9.30 | 80.90 |
| GraphVAE | 55.70 | 76.00 | 61.60 |
| RVAE | 96.60 | - | 97.50 |
| MolGAN | 98.10 | 10.40 | 94.20 |
| GraphNVP | $83.10_{\pm0.50}$ | $99.20_{\pm0.30}$ | $58.20_{\pm1.90}$ |
| GRF | $84.50_{\pm0.70}$ | $66.00_{\pm1.14}$ | $58.60_{\pm0.82}$ |
| GraphAF | 100.00 | 94.51 | 88.83 |
| MoFlow | $100.00_{\pm0.00}$ | $98.53_{\pm0.14}$ | $96.04_{\pm0.10}$ |
| **GraphEBM** | $100.00_{\pm0.00}$ | $97.90_{\pm0.14}$ | $97.01_{\pm0.17}$ |

Table 3: Generation performance on ZINC250k.

| Method | Validity(%) | Uniqueness(%) | Novelty(%) |
|---|---|---|---|
| GCPN | 100.00 | 99.97 | 100.00 |
| JT-VAE | 100.00 | 100.00 | 100.00 |
| MolecularRNN | 100.00 | 99.89 | 100.00 |
| GraphNVP | $42.60_{\pm1.60}$ | $94.80_{\pm0.60}$ | $100.00_{\pm0.00}$ |
| GRF | $73.40_{\pm0.62}$ | $53.7_{\pm2.13}$ | $100.00_{\pm0.00}$ |
| GraphAF | 100.00 | 99.10 | 100.00 |
| MoFlow | $100.00_{\pm0.00}$ | $99.99_{\pm0.01}$ | $100.00_{\pm0.00}$ |
| **GraphEBM** | $99.96_{\pm0.02}$ | $98.79_{\pm0.15}$ | $100.00_{\pm0.00}$ |

Since the dataset for a single property is often available, suppose we have two energy functions $E_{\theta_1^*}(X, A)$ and $E_{\theta_2^*}(X, A)$, which are trained towards two property objectives $O_1$ and $O_2$ respectively, as described in Section 3.6. To generate molecules that have properties $O_1$ and $O_2$ simultaneously, we can obtain a new energy function by summing the above two energy functions since the product of probabilities is equivalent to the sum of corresponding energies, according to Eq. (1). This is known as the product of experts (Hinton, 2002). Formally,

$$p_{\theta_1^*}(X, A) \cdot p_{\theta_2^*}(X, A) = \frac{e^{-E_{\theta_1^*}(X,A)}}{Z(\theta_1^*)} \cdot \frac{e^{-E_{\theta_2^*}(X,A)}}{Z(\theta_2^*)} \propto e^{-\left(E_{\theta_1^*}(X,A)+E_{\theta_2^*}(X,A)\right)}. \tag{9}$$

Nota that $Z(\theta_1^*)$ and $Z(\theta_2^*)$ are constant *w.r.t.* various $(X, A)$. Then we can apply the generation process described in Section 3.5 to $E_{\theta^*}(X, A) = E_{\theta_1^*}(X, A) + E_{\theta_2^*}(X, A)$ to generate molecules towards multiple objectives.

## 4 EXPERIMENTS

### 4.1 SETUP

**Datasets**. We evaluate our proposed method on two widely used molecule datasets, QM9 (Ramakrishnan et al., 2014) and ZINC250k (Irwin et al., 2012). QM9 consists of 134k organic molecules and the maximum number of atoms is 9. It contains 4 atom types and 3 bond types. ZINC250k has 250k drug-like molecules and the maximum number of atoms is 38. It includes 9 atom types and 3 edge types.

**Implementation details**. We kekulize molecules and remove their hydrogen atoms using RD-Kit (Landrum et al., 2006). In our parameterized energy function, we adopt a network of $L = 3$ layers with hidden dimension $d = 64$. We use Swish (Ramachandran et al., 2017) as the activation function. We set the standard variance $\sigma = 0.005$ in the gaussian noise. For training, we tune the following hyperparameters: the scale $t$ of uniform noise $\in [0, 1)$, the sample step $K$ of Langevin dynamics $\in [30, 300]$, and the step size $\frac{\lambda}{2} \in [10, 50]$. All models are trained for up to 20 epochs with a learning rate of 0.0001 and a batch size of 128. We follow the techniques adopted by Du & Mordatch (2019) to stabilize the training process. Specifically, we add spectral normalization (Miyato et al., 2018) to all layers of the network. In addition, we clip the gradient used in Langevin dynamics so that its value magnitude can be less than 0.01. All experiments are run with a single GeForce RTX 2080 Ti GPU.

**Evaluation standards**. We evaluate our proposed method for molecule generation under three settings: random generation, single-objective generation, and multi-objective generation. For random generation, we compare the widely used quantitative metrics with baselines and visualize the generated molecules. For single-objective and multi-objective generation, we show the distribution comparison with random generation in terms of property values. In addition, we perform molecule optimization, including property optimization and constrained property optimization, to further demonstrate the effectiveness of our proposed single-objective generation method. The details of the experimental setup for each setting are included in Appendix F.

### 4.2 RANDOM GENERATION

The quantitative results on QM9 and ZINC250k are shown in Table 2&3. We can observe that GraphEBM performs competitively with existing methods, which is significant considering that the

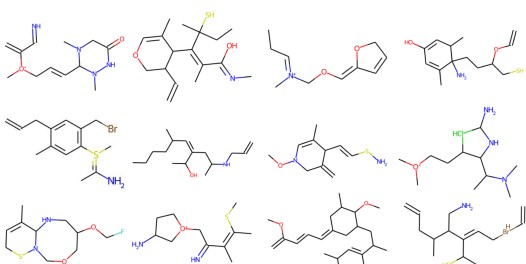

Figure 2: Visualization of generated molecules.

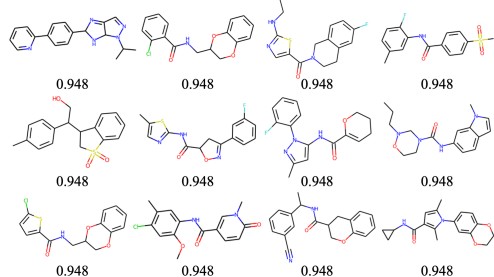

Figure 3: Discovered examples with highest QED scores.

study of EBMs is still in its early stage, compared with other generative models such as GANs and Flows. Generated samples are visualized in Figure 2, which further demonstrates that GraphEBM can generate non-trivial molecules.

To better understand the implicit generation via Langevin dynamics, we visualize this process for an example in Figure 7, Appendix G. We can observe that Langevin dynamics effectively refines the random initialized sample to approach a data point that corresponds to a realistic molecule.

### 4.3 SINGLE-OBJECTIVE GENERATION

Figure 4 (a)&(b) compare the property value distribution between molecules obtained by single-objective generation and random generation. It can be observed that single-objective generation can generate more molecules with high property values, indicating that our proposed strategy for single-objective generation, which aims to assign lower energies to molecules with higher property values, is effective.

The property optimization results are shown in Table 4. Property optimization aims at generating novel molecules with high QED scores. We observe that GraphEBM can find more novel molecules with the best QED score (0.948) than baselines. This strongly demonstrates the effectiveness of our proposed single-objective generation method. Examples of discovered novel molecules with highest QED scores are illustrated in Figure 3.

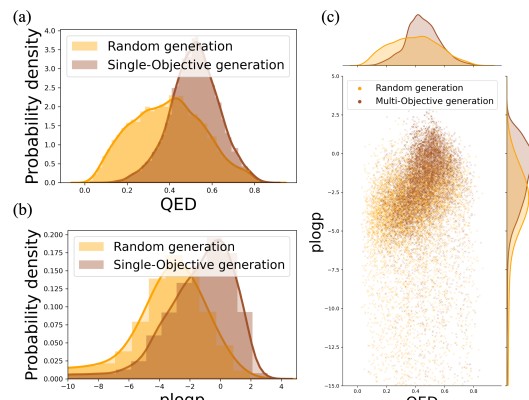

Figure 4: (a)&(b) Comparison of QED and plogp distributions between single-objective generation and random generation, respectively. (c) Comparison of distributions on QED and plogp between multi-objective generation and random generation.

For constrained property optimization, given a molecule, our task is to obtain a new molecule that has a better desired chemical property while preserving similarity. As demonstrated in Table 5, GraphEBM can obtain higher property improvements over JT-VAE, GCPN, and MoFlow by significant margins, and performs competitively with GraphAF. In addition, it can be observed from Shi et al. (2019) that GraphAF learns to improve plogp by simply adding long carbon chains, while our GraphEBM learns more advanced chemical knowledge. Several examples of constrained

Table 4: Property optimization results in terms of the best QED scores. Our GraphEBM finds more novel molecules with the best QED scores.

| Method | 1st | 2nd | 3rd | 4th |
|---|---|---|---|---|
| JT-VAE | 0.925 | 0.911 | 0.910 | - |
| GCPN | 0.948 | 0.947 | 0.946 | - |
| GraphAF | 0.948 | 0.948 | 0.947 | 0.946 |
| MoFlow | 0.948 | 0.948 | 0.948 | 0.948 |
| **GraphEBM** | **0.948** | **0.948** | **0.948** | **0.948** |

property optimization are shown in Figure 5. It is interesting that the modifications are interpretable to some degree. Specifically, in the first example, our model optimizes the plogp score with a remarkable margin of 18.03 by replacing several carbon atoms with sulfur atoms, which could make the molecule more soluble in water, thus leading to a larger logP value. Additionally, plogp is highly

Table 5: Constrained property optimization results. Note that there are two different settings in baselines. JT-VAE and GCPN choose 800 molecules with the lowest plogp scores in the test set and use them as initialization, while GraphAF and MoFlow choose from the training set. We report our results on both of these two settings for extensive comparisons. $\delta$ is the similarity constraint. The top two highest property improvements on each constraint are highlighted as **1st** and 2nd.

| | JT-VAE | | | GCPN | | | GraphEBM | | |
|---|---|---|---|---|---|---|---|---|---|
| $\delta$ | **Improvement** | **Similarity** | **Success** | **Improvement** | **Similarity** | **Success** | **Improvement** | **Similarity** | **Success** |
| 0.0 | $1.91_{\pm 2.04}$ | $0.28_{\pm 0.15}$ | 98% | $\underline{4.20}_{\pm 1.28}$ | $0.32_{\pm 0.12}$ | 100% | $\mathbf{5.76}_{\pm 4.69}$ | $0.08_{\pm 0.13}$ | 98% |
| 0.2 | $1.68_{\pm 1.85}$ | $0.33_{\pm 0.13}$ | 97% | $\mathbf{4.12}_{\pm 1.19}$ | $0.34_{\pm 0.11}$ | 100% | $\underline{3.97}_{\pm 3.77}$ | $0.35_{\pm 0.14}$ | 92% |
| 0.4 | $0.84_{\pm 1.45}$ | $0.51_{\pm 0.10}$ | 84% | $\underline{2.49}_{\pm 1.30}$ | $0.47_{\pm 0.08}$ | 100% | $\mathbf{2.84}_{\pm 3.44}$ | $0.53_{\pm 0.10}$ | 88% |
| 0.6 | $0.21_{\pm 0.71}$ | $0.69_{\pm 0.06}$ | 46% | $\underline{0.79}_{\pm 0.63}$ | $0.68_{\pm 0.08}$ | 100% | $\mathbf{1.52}_{\pm 2.65}$ | $0.68_{\pm 0.05}$ | 64% |

| | GraphAF | | | MoFlow | | | GraphEBM | | |
|---|---|---|---|---|---|---|---|---|---|
| $\delta$ | **Improvement** | **Similarity** | **Success** | **Improvement** | **Similarity** | **Success** | **Improvement** | **Similarity** | **Success** |
| 0.0 | $\underline{13.13}_{\pm 6.89}$ | $0.29_{\pm 0.15}$ | 100% | $8.61_{\pm 5.44}$ | $0.30_{\pm 0.20}$ | 99% | $\mathbf{15.75}_{\pm 7.40}$ | $0.01_{\pm 0.04}$ | 99% |
| 0.2 | $\mathbf{11.90}_{\pm 6.86}$ | $0.33_{\pm 0.12}$ | 100% | $7.06_{\pm 5.04}$ | $0.43_{\pm 0.20}$ | 97% | $\underline{8.40}_{\pm 6.38}$ | $0.35_{\pm 0.15}$ | 94% |
| 0.4 | $\mathbf{8.21}_{\pm 6.51}$ | $0.49_{\pm 0.09}$ | 100% | $4.71_{\pm 4.55}$ | $0.61_{\pm 0.18}$ | 86% | $\underline{4.95}_{\pm 5.90}$ | $0.54_{\pm 0.11}$ | 79% |
| 0.6 | $\mathbf{4.98}_{\pm 6.49}$ | $0.66_{\pm 0.05}$ | 97% | $2.10_{\pm 2.86}$ | $0.79_{\pm 0.14}$ | 58% | $\underline{3.15}_{\pm 5.08}$ | $0.67_{\pm 0.06}$ | 45% |

Figure 5: Examples of constrained property optimization. The values above and below arrows denote the similarity scores and improvements, respectively. The modifications are highlighted with red rectangles.

related to the number of long cycles and synthetic accessibility. As shown in the second and third examples, our model improves the synthetic accessibility and reduces the number of long cycles by removing or breaking long cycles. These facts indicate that our single-objective generation method can explore the underlying chemical knowledge related to the corresponding property objective.

### 4.4 MULTI-OBJECTIVE GENERATION

The comparison of the distributions on QED and plogp between molecules obtained by multi-objective generation and random generation is illustrated in Figure 4 (c). We can observe that multi-objective generation tends to generate more molecules with both high QED and plogp scores. Additionally, the distribution of QED or plogp is similar to the corresponding distribution obtained by single-objective generation towards a single objective (Figure 4 (a)&(b)). These facts demonstrate that our GraphEBM is able to generate molecules towards multiple objectives in a compositional manner, which provides a novel and promising way for multi-objective molecule generation.

## 5 CONCLUSION

In this paper, we propose GraphEBM as an exploratory work towards permutation invariant and multi-objective molecule generation. GraphEBM is the first EBM for generating molecular graphs with defining a permutation invariant distribution over molecular graphs. For objective-directed generation, we propose to flexibly push down energies according to given property values and further explore to generate molecules towards multiple objectives in a compositional manner, leading to a promising method for multi-objective molecule generation. Experimental results demonstrate that our GraphEBM can generate realistic molecules and the proposals of single-objective and multi-objective generation are effective and promising.

Since EBMs have unique advantages for molecular graph generation, including preserving permutation invariance and enabling multi-objective generation via compositionality, and have rarely been explored for graph generation, we hope our exploratory work would open the door for future research in this area.

## REPRODUCIBILITY STATEMENT

A lot of efforts have been made to ensure the reproducibility of our study. To make our methodology as clear as possible, in addition to the description in Section 3, we include more explanations, discussions, and illustrations in Appendix A, B, C, D, & E. In terms of experiments, we provide the details of our experimental settings, including random, single-objective, and multi-objective generation, in Appendix F. In addition, we describe how our hyperparameters are determined in Section 4.1. We will open-source our implementations once the paper is published.

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

## A  THE DETAILED ANNOTATION OF FIGURE 1

The left part and right part illustrate the processes of obtaining the positive sample and the hallucinated sample, respectively. The middle part shows the forward and backward propagation of the training process. Green and purple arrows represent the forward computation of energy value for the positive sample and the hallucinated sample, respectively. The black dashed arrow denotes the gradient backpropagation. In this example, $n = 9$, $k = 8$, $b = 4$, and $c = 3$. The annotations of node representation matrices denote the atom types, including carbon (C), nitrogen (N), oxygen (O), fluorine (F), and virtual atom ($\star$). Note that we remove hydrogen atoms, which is a common technique in the community. The annotations of adjacency tensor indicate the bond types, including single (S), double (D), triple (T), and virtual bond ($\star$).

## B  MORE DISCUSSION ABOUT DEQUANTIZATION

Note that the dequantization for positive samples is optional. This indicates that we can set $t = 0$ and keep the positive data discrete since Langevin dynamics is only required for obtaining hallucinated samples. Applying dequantization to positive samples can be viewed as a data augmentation technique and we can easily convert the dequantized continuous data back to the original one-hot discrete data by simply applying the argmax operation.

## C  LEARNING ALGORITHM

---

**Algorithm 1** Learning algorithm of GraphEBM

---

**Input:** Observed molecular graph dataset $\mathcal{D} = \left\{(X, A)^{(m)}\right\}_{m=1}^{|\mathcal{D}|}$ with $n$, $b$, and $c$, parameterized energy function $E_\theta(\cdot)$, step size $\frac{\lambda}{2}$, variance of noise $\sigma^2$, number of steps $K$, scaling hyperparameter $t$, weight $\alpha \in \mathbb{R}$ for regularization term

1: **for** $(X, A) \sim \mathcal{D}$ **do**                   ▷ Batch training is applied in practice
2:   $X^\oplus = X + tu$, $u \sim [0, 1)^{n \times (b+1)}$;  $A' = A + tu$, $u \sim [0, 1)^{n \times n \times (c+1)}$   ▷ Eq. (5)
3:   $A^\oplus_{(:,:,k)} = D^{-1}A'_{(:,:,k)}$, $k = 1, \cdots, c + 1$, where $D_{(i,i)} = \sum_{j,k} A'_{(i,j,k)}$   ▷ Eq. (6)
4:   Compute $E_\theta(X^\oplus, A^\oplus)$
5:   $X^0 \sim [0, 1+t)^{n \times (b+1)}$,  $A^0 \sim [0, 1)^{n \times n \times (c+1)}$
6:   **for** sample step $k = 1$ to $K$ **do**                   ▷ Eq. (4)
7:     $X^k = X^{k-1} - \frac{\lambda}{2}\nabla_X E_\theta\left(X^{k-1}, A^{k-1}\right) + w^k$,  $w^k \sim \mathcal{N}(0, \sigma^2)$
8:     $A^k = A^{k-1} - \frac{\lambda}{2}\nabla_A E_\theta\left(X^{k-1}, A^{k-1}\right) + \eta^k$,  $\eta^k \sim \mathcal{N}(0, \sigma^2)$
9:     Clamp $X^k$ and $A^k$ such that the values are in the desirable ranges
10:   **end for**
11:   $X^\odot = X^K$, $A^\odot = A^K$
12:   Compute $E_\theta(X^\odot, A^\odot)$
13:   Compute $\mathcal{L}_{energy} = E_\theta(X^\oplus, A^\oplus) - E_\theta(X^\odot, A^\odot)$          ▷ Eq. (7)
14:   Compute $\mathcal{L}_{reg} = E_\theta(X^\oplus, A^\oplus)^2 + E_\theta(X^\odot, A^\odot)^2$ ▷ Regularizer is used for stable training
15:   Compute $\mathcal{L} = \mathcal{L}_{energy} + \alpha\mathcal{L}_{reg}$               ▷ Total loss function
16:   Update $\theta$ based on $\nabla_\theta \mathcal{L}$
17: **end for**

---

# D   AN ILLUSTRATION OF GENERATION PROCESS

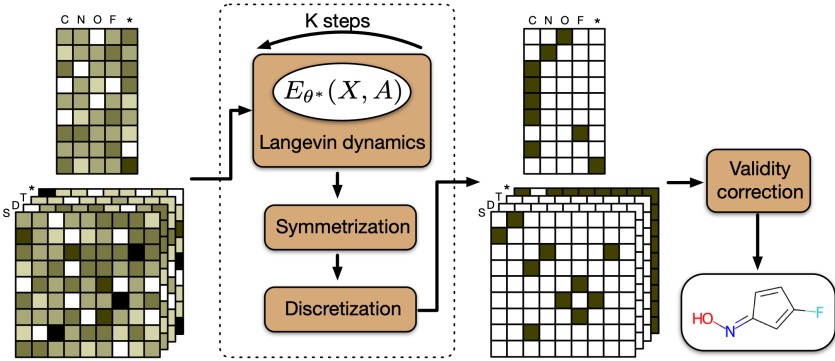

Figure 6: The generation process of our GraphEBM.

# E   EXISTING SINGLE-OBJECTIVE GENERATION STRATEGIES

There are mainly three approaches in the literature for single-objective generation. First, this task can be modeled as a conditional generation problem, where the property value can be utilized as the condition (Simonovsky & Komodakis, 2018). Second, for methods using the latent space, a predictor can be applied to learn the property value from the latent representation (Gómez-Bombarelli et al., 2018). Third, reinforcement learning can be used to optimize the properties of generated molecules (You et al., 2018). However, it is not straightforward to apply these methods to our GraphEBM for single-objective generation since GraphEBM generates molecules implicitly using Langevin dynamics and no latent space exists. Also, using EBMs for generation that is conditional on continuous conditions is rarely studied by the community. Hence, it remains challenging to apply EBMs for single-objective generation.

# F   EXPERIMENTAL SETUP FOR EACH SETTING

**Random generation**. We evaluate the ability of our proposed GraphEBM to model and generate molecules. We consider most methods reviewed in Section 2.1 as baselines. The following commonly used metrics are adopted. *Validity* is the percentage of chemically valid molecules among all generated molecules. *Uniqueness* denotes the percentage of unique molecules among all valid molecules. *Novelty* corresponds to the percentage of generated valid molecules that are not present in the training set. The metrics are computed on $10,000$ randomly generated molecules. Results averaged over $5$ runs are reported in Table 2&3. For QM9, the results of CVAE and GVAE are obtained from Simonovsky & Komodakis (2018). The result of MoFlow is obtained by evaluating its public trained model. All other results are from their original papers. For ZINC250k, the results of GCPN and JT-VAE are obtained from Shi et al. (2019). The result of MoFlow is obtained by evaluating its public trained model. All other results are from their original papers.

**Single-Objective generation**. To empirically show the effectiveness of our single-objective generation method proposed in Section 3.6, we train models on ZINC250k accordingly and compare the distribution of the property score between molecules obtained by single-objective generation and random generation. We consider two chemical properties, including Quantitative Estimate of Druglikeness (QED) (Bickerton et al., 2012) and penalized logP (plogp), which is the water-octanol partition coefficient penalized by the number of long cycles and synthetic accessibility.

We further verify the effectiveness of our proposed single-objective generation method by performing molecule optimization, including property optimization and constrained property optimization. Property optimization aims at generating novel molecules with high QED scores. We directly use the model trained for single-objective generation and leverage the molecules in the training set as initialization for Langevin dynamics, following prior works (Jin et al., 2018; Zang & Wang, 2020). We report the highest QED scores and the corresponding novel molecules discovered by our method.

For constrained property optimization, given a molecule $m$, our task is to obtain a new molecule $m'$ that has a better desired chemical property with the molecular similarity $sim(m, m') \geq \delta$ for some threshold $\delta$. We adopt Tanimoto similarity of Morgan fingerprint (Rogers & Hahn, 2010) to measure the similarity between molecules. We find that there are two different settings in baselines. JT-VAE and GCPN choose $800$ molecules with the lowest plogp scores in the test set and use them as initialization, while GraphAF and MoFlow choose from the training set. We report our results on both of these two settings for extensive comparisons.

**Multi-Objective generation**. As investigated in Section 3.7, our GraphEBM has the potential to conduct compositional generation towards multiple objectives. To verify this, we combine the two energy functions obtained in single-objective generation experiments, as formulated in Eq. (9). Then we apply the generation process described in Section 3.5 to the resulting energy function to generate molecules. To show the effectiveness of our proposal, we compare the distribution of the property scores between molecules generated by multi-objective generation and random generation.

# G VISUALIZATION OF THE IMPLICIT GENERATION PROCESS

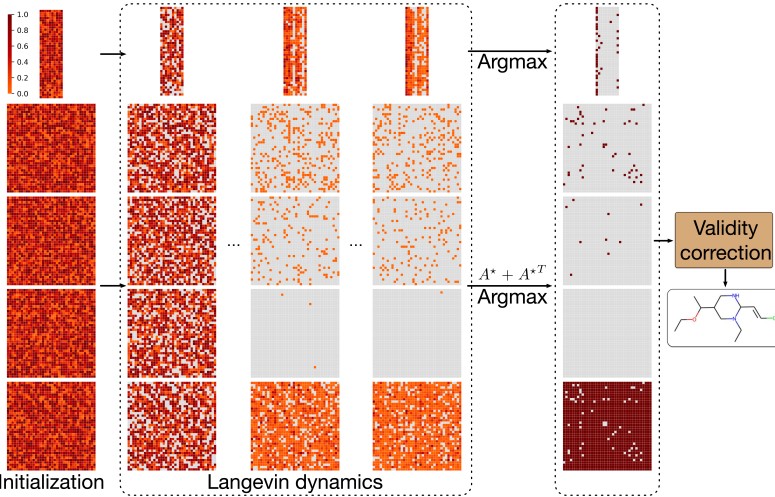

Figure 7: Visualization of the implicit generation process of our GraphEBM. The first row denotes atom matrices and the remaining rows represent fours channels of adjacency tensors, corresponding to single, double, triple, and virtual bonds. For better visual results, each atom matrix and adjacency tensor is normalized by dividing by its maximum value.

