# OpenReview forum: "GraphEBM: Towards Permutation Invariant and Multi-Objective Molecular Graph Generation"
_ICLR.cc/2022/Conference — ICLR 2022 Submitted_

### Official Review · Reviewer_JCco · 2021-10-27

**Correctness:** 3
**Technical Novelty And Significance:** 2
**Empirical Novelty And Significance:** 3
**Recommendation:** 3
**Confidence:** 3

**Main Review:**

Strength
1. This paper propose an energy-based generative model for molecule generation, which is a novel contribution of molecule generation literature.

2. This paper is well-written, and easy to read.

Weakness
1. The discussion on the main contribution of single objective optimization (Eq. 8) lacks. The proposed modification scale up the positive energy more than 1, then it is not MLE solution of Boltzman distribution modeled by the energy function. I know this modification practically works well as in the experiments, but this should be discussed in the paper.

2. I am not expert of molecule generation, but the improvement of proposed method is too marginal. For example, the Table 4 does not show any meaningful results; and in the Table 5 the proposed method only shows better performance on the most relaxed constraints setting that allows any modification (delta=0.0).

**Summary Of The Paper:**

This paper proposed a graph energy-based model based for molecular graph generation. The energy function is implemented by graph neural networks, and a molecule is generated by the Langevin dynamics that maximizes the trained energy function. Further, the author proposed to re-weight energy by a function of property value for the single objective optimization, and to sum up energy functions trained on different property datasets for the multiple objective optimization. The experimental results show the effectiveness of the proposed method on benchmark molecular graph generation tasks.

**Summary Of The Review:**

This paper propose a novel method of energy-based model for molecule generation implemented by GNN. However, the modification of loss function is not justified properly and the improvement is too marginal.

---

### Official Review · Reviewer_Fo2L · 2021-11-01

**Correctness:** 4
**Technical Novelty And Significance:** 3
**Empirical Novelty And Significance:** 2
**Recommendation:** 5
**Confidence:** 4

**Main Review:**

Overall, the paper is well-written. I like the simplicity of the proposal and the pursuit of permutation invariant generation. Notably, this seems to be the first proposal of EBM-based generative models for molecules and attributed graphs. I also agree with the authors that achieving multi-objective property optimization via the compositionality of energy functions is an interesting aspect of EBMs.

My concerns/questions mainly refer to the degree of novelty (C1), simplified evaluation setup (C2), lack of performance gains (C3), ability to generate valid molecules without posthoc correction (C4), and better motivation for some modeling choices (C5).

- C1: Some of the contributions listed by the authors (in Section 1) derive naturally from the use of EBMs. For instance, the compositional generative aspect of EBMs has been explored for image data in [1]. Therefore, it seems like the paper's main contribution is to use GNNs to parameterize energy functions.

- C2: The multi-objective setup only includes the QED and LogP properties, which are highly correlated. It would be important to consider more realistic scenarios with different targets, such as those used in [2].

- C3: The method only achieves competitive results and doesn't provide significant gains over the baselines. For instance, GraphEBM is not the best performing method in any of the metrics in Tables 2 and 3. Also, in Table 5, the proposed GraphEBM reports higher variances than GCPN.

- C4: The authors should report the percentage of generated valid molecules before applying the posthoc correction procedure. This would allow us to assess the quality of the generative process via SGLD.

- C5: What is the rationale of choosing $f(y)=1+e^{y}$, where $y \in [0, 1]$? This seems to significantly limit how much the model pushes down the energy of favorable molecules.

**Minor issue**: I would not say that "...the study of EBMs is still in its early stage compared with other generative models such as GANs and Flows" as the appearance of EBMs dates to the '80s.

Based on the issues above, my recommendation is marginally below the acceptance threshold. As usual, I am open to increasing my score depending on the authors' rebuttal.

[1] Du & Mordatch. Implicit Generation and Modeling with Energy-Based Models. NeurIPS, 2019.

[2] Jin et al., Multi-Objective Molecule Generation using Interpretable Substructures. ICML, 2020.

**Summary Of The Paper:**

The paper proposes a new energy-based generative model for molecules. The idea is straightforward and relies on using R-GNNs (relational graph neural networks) to parameterize the energy function. The authors show how to achieve single- and multi-objective generation via scaling, and composition, respectively. The experiments mostly follow the setup introduced in JT-VAE (Junction-Tree VAEs), and the results show the proposal is competitive against existing models.

**Summary Of The Review:**

The main positive aspects of this paper are: $i$) first EBM-based model for molecules; $ii$) a model that leverages the node permutation-invariant nature of graphs/molecules. On the other hand, the main negative aspects include $i$) limited evaluation setup; $ii$) no significant improvements over baselines (the empirical gains are questionable).

---

### Official Review · Reviewer_EHKx · 2021-11-02

**Correctness:** 4
**Technical Novelty And Significance:** 3
**Empirical Novelty And Significance:** 2
**Recommendation:** 6
**Confidence:** 3

**Main Review:**

Strength: as the auther claim, this is the very first paper that proposes to use energy-based models to generate molecule graphs. It is well written and experimental results look promising.

weakness: As the main goal of the paper was to overcome the challenge of permutation invariance property when we generate graphs, it would be good to have a discussion in the paper the computational cost of the proposed method when compared to other baselines that use different tricks to handle the permutation invariance.

I also wonder about the sampling cost of this method, as it applies Langevin dynamics to generate samples, what is the time cost and how many iteration it normally requires?

**Summary Of The Paper:**

The paper proposes an energy-based molecular graph generative model. By using a permutation invariant energy function, the model is able to model a permutation invariant distribution over molecular graphs, thus preserving the intrinsic permutation invariance
property during density modeling. They propose to use contrastive divergence to learn the energy function and generate samples from it via Langevin dynamics. They also proposed goal-directed generation where they push down the energy with a flexible degree according to the property value of desirable properties.

**Summary Of The Review:**

The paper is well written and the presenting idea seems interesting and novel.

---

### Official Review · Reviewer_tnpH · 2021-11-03

**Correctness:** 3
**Technical Novelty And Significance:** 3
**Empirical Novelty And Significance:** 3
**Recommendation:** 3
**Confidence:** 4

**Main Review:**

Pros：

- This paper proposes to use EBMs for task-based molecule generation, which is exploring a different direction from the most popular autoregressive-based generation.

- The paper is well-written and easy to follow.

- Experiments are comprehensive in the sense that the authors conduct both single-objective optimization, constrained optimization, and multi-objective optimization to verify the proposed method.

Cons:

- The idea of using EBMs in this paper is somewhat straightforward and incremental. From my understanding, one of the most technically difficult parts of using EBMs on graphs lies in the generation, from continuous domain to discrete domain. However, this paper applies simple post-processing and does not address any of the issues here. It would be more helpful to justify such post-processing using theoretical or empirical evidence.

- In the constrained property optimization experiment, the authors only show the results without mentioning how to use their method for tasks with structure constraints.

- The paper shows many experiments, but it is not clear which one is the main experiment. In previous work, the harder task is to generate molecules with multiple (4) properties[1,2,3]. In the current paper, the single-objective optimization seems to be a sanity check. It is unclear how to do constrained property optimization. And it is hard to quantify the multi-objective results.

- Writing issue: the paper puts too much emphasis on the permutation invariance property of the proposed method. Admittedly, the proposed method has the advantage of permutation invariance compared with autoregressive generative models because it does not rely on conditional generation and the energy function itself is a permutation invariant GCN. However, there is no necessity to spend the entire subsection 3.3 on this. The paper would be more persuasive if the authors can demonstrate this benefit directly using empirical results.


Question:

- The authors propose to use sum pooling for aggregating the graph level information. Would this result in different scaling when computing the energy? i.e., large molecule will be more likely to result in larger energy values?

- How does summing two energy functions compared with using the geometric mean of two objective scores and train only one energy function? Does training one extra energy function helps generate better molecules?


[1] Li, Yibo, Liangren Zhang, and Zhenming Liu. "Multi-objective de novo drug design with conditional graph generative model." Journal of cheminformatics 10.1 (2018): 1-24.
[2] Jin, Wengong, Regina Barzilay, and Tommi Jaakkola. "Multi-objective molecule generation using interpretable substructures." International Conference on Machine Learning. PMLR, 2020.
[3] Chen, Binghong, et al. "Molecule Optimization by Explainable Evolution." International Conference on Learning Representations. 2021.

**Summary Of The Paper:**

The paper proposes to use energy-based models (EBMs) for task-based molecule generation. The energy function is parameterized using relational graph convolutional networks (R-GCN) which have the permutation invariance property. In order to learn an energy landscape that can generate high-scoring molecules, the authors propose to weigh the positive samples by their objective scores during the contrastive divergence training and to combine several energy functions for multi-objective molecule generation. Experiments on single- and multi-objective optimization tasks demonstrate its ability to generate molecules with high scores.

**Summary Of The Review:**

Overall, due to the weakness in novelty and experiments of this paper, I recommend rejecting it in its current form. But I encourage the authors to improve the paper according to the comments and resubmit it in the future.

---

> ### Comment · Reviewer_tnpH · 2021-11-29
> **Finalized review**
>
> I agree with reviewers Fo2L and JCco on limited novelty, simplified experiment setups, and marginal performance gain. Therefore I would keep my original score and recommend rejecting the paper.

---

### Decision · Program_Chairs · 2022-01-20

**Decision:**

Reject

**Comment:**

This paper proposes to use an energy-based model for a multi-objective molecular generation. The energy function is parameterized by relational graph convolutional network (R-GCN) so that it has a permutation invariance property. The model is trained by contrastive divergence and the generation is performed by Langevin dynamics. Experiments on single and multi-objective molecule generation are conducted to verify the effectiveness of the proposed framework. The paper is well-written, and the experiments are comprehensive. The major shortcoming of the paper is its limited novelty, since using EBM for graph generation is a straightforward application of the existing deep EBM framework. The contribution is marginal.

During the discussion, two of the reviewers pointed out that the contribution is limited and marginal. Two reviewers pointed out that the performance gain obtained by the proposed model is marginal and not significant. One reviewer has a concern about the computational cost of MCMC. However, the authors didn’t provide a rebuttal to address the concerns raised by the reviewers. Given the fact that all the concerns from the reviewers remain, and the contribution and performance gain of the work are marginal, the AC recommends rejecting the paper.